# Evidence for an adverse impact of remote readouts on radiology resident productivity: Implications for training and clinical practice

**Emile B. Gordon**[1], **Peter Wingrove**[1], **Barton F. Branstetter IV**[1,2], **Marion A. Hughes**[1,2]*

**1** Department of Radiology, University of Pittsburgh Medical Center, Pittsburgh, Pennsylvania, United States of America, **2** Department of Otolaryngology, University of Pittsburgh Medical Center, Pittsburgh, Pennsylvania, United States of America

* hughesma@upmc.edu

**Data Availability Statement:** On the Harvard Dataverse repository, at this link: https://doi.org/10.7910/DVN/MHGZWL.

## Abstract

After their rapid adoption at the onset of the coronavirus pandemic, remote case reviews (remote readouts) between diagnostic radiology residents and their attendings have persisted in an increasingly remote workforce, despite relaxing social distancing guidelines. Our objective was to evaluate the impact of the transition to remote readouts on resident case volumes after the recovery of institutional volumes. We tabulated radiology reports co-authored by first-to-third-year radiology residents (R1-R3) between July 1 and December 31 of the first pandemic year, 2020, and compared to the prior two pre-pandemic years. Half-years were analyzed because institutional volumes recovered by July 2020. Resident volumes were normalized to rotations, which were in divisions categorized by the location of the supervising faculty during the pandemic period; in 'remote' divisions, all faculty worked off-site, whereas 'hybrid' divisions had a mix of attendings working on-site and remotely. All residents worked on-site. Data analysis was performed with Student's *t* test and multivariate linear regression. The largest drops in total case volume occurred in the two remote divisions (38% [6,086 to 3,788], and 26% [11,046 to 8,149]). None of the hybrid divisions with both in-person and remote supervision decreased by more than 5%. With multivariate regression, a resident assigned to a standardized remote rotation in 2020 would complete 32% (253 to 172) fewer studies than in identical pre-pandemic rotations (coefficent of −81.6, $p = .005$) but would be similar for hybrid rotations. R1 residents would be expected to interpret 40% fewer (180 to 108) cases on remote rotations during the pandemic (coefficient of −72.3, $p = .007$). No significant effect was seen for R2 or R3 residents ($p = .099$ and $p = .29$, respectively). Radiology residents interpreted fewer studies during remote rotations than on hybrid rotations that included in-person readouts. As resident case volume is correlated with clinical performance and board pass rate, monitoring the readout model for downstream educational effects is essential. Until evidence shows that educational outcomes remain unchanged, radiology residencies may wish to preserve in-person resident readouts, particularly for junior residents.

**Funding:** The authors received no specific funding for this work.

**Competing interests:** The authors have declared that no competing interests exist.

## Author summary

The rapid adoption and continued prevalence of remote work following the coronavirus-19 pandemic has profoundly impacted medical practice and resident training. The shift has particularly affected diagnostic radiology, which is amenable to remote work but is also traditionally centered around in-person, workstation-based case reviews (readouts) between residents and attendings. However, the impact of remote readouts on resident productivity remains largely unexplored. Our study investigates resident case volumes during the pandemic, after the recovery of institutional volumes (July to December 2020) and compares the volumes to same period in three preceding pre-pandemic years. Our study shows a substantial decrease in resident case volumes when working in divisions where attending faculty worked remotely, with the most pronounced impact on first-year residents. On the other hand, 'hybrid' divisions, with a mix of on-site and off-site attendings, maintained similar resident case volumes to pre-pandemic levels. Given the correlation of case volumes with clinical and academic performance, our findings suggest monitoring for unintended consequences on radiology resident educational outcomes.

## Introduction

In-person radiology case review, also known as a readout, by a supervising attending radiologist is a time-honored, integral component of radiology resident education [1]. When the World Health Organization declared the coronavirus disease 2019 (COVID-19) pandemic a global emergency in March 2020 [2], this changed rapidly, as government shutdowns and mandatory social isolation necessitated a shift from in-person to remote readouts [3,4]. In response to local, state, and federal mandates, there was an immediate reduction in procedural and imaging case volume [5–7] attributable to the cancellation of elective studies and reduction of inpatient and emergency imaging [8,9]. This decrease in case volumes was mirrored in the radiology resident experience, with a nearly two-thirds decline in resident-interpreted studies [5,10].

The interim reduction of case volumes was likely detrimental to trainee education, as case volumes are known to correlate with radiology resident performance in the reading room and on board exams [11,12]. However, while institutional volumes have since normalized, remote readouts have persisted in order to comply with social distancing regulations and to address radiology staffing needs; furthermore, as part of a broader shift towards telemedicine throughout medical practice [13], remote readouts are expected to continue in the future [14,15].

While there is evidence that the transition to a remote workplace during the pandemic may have benefited didactic education and conferences [16,17], and that many residents and faculty may support this model [16,18], the impact of remote readouts of resident case volumes is still largely unknown. Secondary outcome data by Poyiadji et al. 2020 suggest lagging resident volumes despite the normalization of faculty-only volumes, whereas there are conflicting early data regarding resident case volumes when reading out on at-home versus on-site workstations [19,20]. However, the results of the recent 2021–2022 survey by the American Alliance of Academic Chief Residents in Radiology reveal an overwhelming resident preference for in-person readouts, case conferences, and didactics [21].

The purpose of this study is to evaluate the impact of the transition to remote readouts on resident case volumes after the recovery of institutional volumes.

## Materials and methods

### Data acquisition

A retrospective analysis of the number of studies co-authored by radiology residents in seven divisions within the institution's academic core was performed for six months from July to December 2020, during which institutional imaging volumes recovered after the initial decline of volumes during the early pandemic. Comparison data from 2018–2019 were obtained during the corresponding six-month periods in those years. Resident training level (R1-R3) was assigned by comparing the imaging study completion year to the expected graduation year (*e. g.*, a PGY-2 trainee is a first-year radiology resident and denoted as R1). Fourth-year radiology residents (R4) were excluded from the analysis because they have highly individualized and variable schedules, as the rotations in the R4 year consist entirely of electives at our institution. For example, many R4 residents at our institution are scheduled for multiple advanced procedural rotations, pursue extended periods of research, and complete non-interpretive rotations such as informatics, 3D printing, and clinical electives within departments outside of radiology. As such, there is tremendous variability in total case volumes dependent on which electives the individual R4 residents choose.

Studies co-authored by residents in the academic core were assigned to one of seven divisions based on the affiliation of the signing attending: abdominal imaging, thoracic imaging, musculoskeletal, neuroradiology, nuclear medicine, pediatric radiology, and women's imaging. Divisions were classified as employing a remote or hybrid readout model based on the location of the supervising faculty from July to December 2020. Faculty of remote divisions were completely off-site during this period, supervising on-site resident readouts remotely. Hybrid divisions had a proportion of the division's attendings consistently in the reading room, available for in-person readouts with residents, and a proportion of attendings off-site, available for remote readouts. The institutional work relative value units (wRVUs)—a standardized billing parameter to estimate physician productivity—for the latter half of 2018, 2019, and 2020 were acquired for comparison as a proxy for changes in overall division volumes. Each study was grouped by modality: computed tomography/magnetic resonance (CT/MR) imaging, fluoroscopy (FL), mammography (MG), nuclear medicine (NM), radiography (XR), or ultrasonography (US). A few studies could not be assigned to any of these modality groups and were omitted.

### Analysis

To evaluate the effect of the pandemic on resident case volumes, the number and type of studies co-authored by residents were analyzed before the pandemic as well as during the pandemic once institutional volumes recovered to normal levels. For the pre-pandemic comparison period, data from July 1st to December 31st of both 2018 and 2019 were utilized to increase statistical power. To compare the rate at which studies were being completed, data on individual imaging studies were aggregated to the level of the resident/division/year as a proxy for a standardized rotation. Although possible for a resident to have completed two separate rotations in the same division within a six-month window, this was rarely observed in our dataset and occurred at similar rates in all years during both pre-pandemic and pandemic periods. Each rotation lasted 3–4 weeks.

The frequency of different modalities within a rotation was tabulated as a percentage of the total volume. Univariate tests for significance were performed using two-way *t* tests. Multivariate linear regression was performed at the rotation level with clustering to account for multiple rotations being completed by each resident. Interaction variables were constructed to examine the differential effect of the pre-pandemic and pandemic periods by readout model. To control

for imaging modality, we utilized two different sets of control variables, a main specification with variables for each modality's frequency as a percentage of total interpreted studies on that rotation, along with an alternative specification that uses categorical variables for each division. The main specification of the regression was also run separately for each resident level to assess for a potential differential impact of the transition to remote readouts on less experienced residents. Values of $p$ less than .05 were considered statistically significant. Mean volumes are reported with 95% confidence intervals (CI). Statistical analysis was conducted in Stata 14.2.

### Ethics approval

The Institutional Review Board reviewed and determined that this study met the regulatory requirements for exempt research.

## Results

During the six-month period from July 1 to December 31, between the years 2018–2020, there were a total of 128,497 studies interpreted by R1, R2, and R3 residents on core academic rotations. After 310 studies of unclear image modality were excluded, the analysis included 128,187 studies interpreted by 56 unique residents (Table 1).

Across all three half-years, the residents partook in a total of 565 rotations. Fourteen of these rotations were excluded because they had 10 or fewer studies; these were most likely delayed reports, addenda from a January to June rotation, or otherwise not representative of a true rotation. This left 551 resident rotations across all the months studied.

All divisions utilized conventional in-person readouts during 2018 and 2019. Two of the seven divisions—thoracic imaging and pediatric radiology—transitioned to a remote readout model during the six-month period, with the faculty reading from offsite workstations. The other five divisions employed a hybrid readout model, with a proportion of attendings consistently available on-site. All the residents were located in onsite reading rooms during the period analyzed.

After the initial decline during the pandemic, our institution's volumes in 2020 returned to near normal between the months of July and December 2020. Departmental wRVUs in 2020 (463,781) were 97% of the 2018 and 2019 average (477,179). Notably, there was a minimal impact of total wRVUs for the remote divisions, with pediatric and thoracic radiology interpreting 98% (52,360/53,516) and 102% (31,207/30,697) of their previous wRVUs. At our institution, the attending radiologists interpret the majority of imaging studies (>50%) without residents, and therefore the small variation in total studies from prior years likely had a negligible effect on resident case volumes.

### Total resident case volumes

The number of studies co-authored by R1-R3 residents during the pre-pandemic half-years of 2018 (N = 44,607) and 2019 (N = 44,391) was nearly identical, whereas the latter half of 2020 exhibited a 12% decrease from the average previously (39,189/44,499) (Table 1). However, there was substantial variability in resident level, division readout model (hybrid versus remote), and modality. Rotations in divisions adopting a hybrid readout approach remained essentially unchanged (27,252/27,368; −0.4%), while rotations in divisions adopting only remote readouts during the pandemic saw a decrease of more than 30% (11,937/17,131). The two divisions that conducted only remote readouts during the pandemic had the largest decreases in total resident volume; the thoracic imaging rotations saw a 38% decrease in total resident interpreted studies from the average previously (3,788/6,086), and pediatric imaging rotations had a 26% decrease (8,149/11,046). By contrast, none of the hybrid division rotations

**Table 1. Characterization of the number of residents and resident co-authored studies.**

| | All years | Pre-pandemic | | | Pandemic | % change ‡ |
| | | 2018* | 2019* | Average | 2020* | |
|---|---|---|---|---|---|---|
| No. Residents † | 56 | 32 | 33 | 32.5 | 33 | 1.5% |
| *No. residents by level* | | | | | | |
| R1 | 33 | 10 | 11 | 10.5 | 12 | 14.3% |
| R2 | 32 | 11 | 11 | 11 | 10 | -9.1% |
| R3 | 33 | 11 | 11 | 11 | 11 | 0.0% |
| No. studies | 128,187 | 44,607 | 44,391 | 44,499 | 39,189 | -11.9% |
| *Resident level* | | | | | | |
| R1 | 42,558 | 13,875 | 15,034 | 14,455 | 13,649 | -5.6% |
| R2 | 42,064 | 16,201 | 15,519 | 15,860 | 10,344 | -34.8% |
| R3 | 43,565 | 14,531 | 13,838 | 14,185 | 15,196 | 7.1% |
| *Readout model* | | | | | | |
| Hybrid | 81,988 | 27,513 | 27,223 | 27,368 | 27,252 | -0.4% |
| Remote | 46,199 | 17,094 | 17,168 | 17,131 | 11,937 | -30.3% |
| *Division* | | | | | | |
| AI | 33,991 | 11,296 | 11,378 | 11,337 | 11,317 | -0.2% |
| Thoracic | 15,959 | 6,231 | 5,940 | 6,086 | 3,788 | -37.8% |
| MSK | 14,723 | 5,298 | 4,615 | 4,957 | 4,710 | -5.0% |
| Neuro | 19,681 | 6,323 | 6,905 | 6,614 | 6,453 | -2.4% |
| NM | 6,744 | 2,488 | 1,766 | 2,127 | 2,490 | 17.1% |
| Pediatric | 30,240 | 10,863 | 11,228 | 11,046 | 8,149 | -26.2% |
| Women's | 6,849 | 2,008 | 2,559 | 2,284 | 2,282 | -0.1% |
| *Modality* | | | | | | |
| CT/MR | 38,443 | 11,856 | 13,492 | 12,674 | 13,095 | 3.3% |
| FL | 5,002 | 1,601 | 1,713 | 1,657 | 1,688 | 1.9% |
| MG | 3,666 | 1,143 | 1,322 | 1,233 | 1,201 | -2.6% |
| NM | 5,855 | 2,207 | 1,600 | 1,904 | 2,048 | 7.6% |
| XR | 55,254 | 20,815 | 19,727 | 20,271 | 14,712 | -27.4% |
| US | 19,967 | 6,985 | 6,537 | 6,761 | 6,445 | -4.7% |

† R1 to R3 residents

*July 1st through December 31st of these years

‡ Percent change of the average number of studies during the latter half of 2018 and 2019 (N/N̄) to 2020. AI: abdominal imaging; Thoracic: thoracic imaging; MSK: musculoskeletal; Neuro: neuroradiology; NM: nuclear medicine; Pediatric: pediatric radiology. FL: Fluoro; MG: Mammography; XR: Radiographs; US: Ultrasound, NM: Nuclear imaging, CT/MR: Cross-sectional studies

saw a decrease of more than 5% (see Table 1). By imaging modality, there were 27% (14,712/20,271) fewer radiographs and 8% (2,048/1,904) more nuclear medicine studies interpreted by residents during the pandemic period, with no other modality changing by more than 5%.

## Mean resident case volume per rotation before and during the pandemic

After normalizing to mean resident case volume per rotation, *t* tests demonstrated no significant difference ($p = .13$) in the overall mean number of studies between pre-pandemic rotations (239.6 CI [222.2, 260.0]) and pandemic rotations (215.3 CI [190.8, 239.8]) (Table 2). However, when looking at rotations in divisions that transitioned to a remote readout model during the pandemic period (pediatric and thoracic divisions combined), there was a statistically significant ($p = 0.043$) decrease in the mean number of studies interpreted by each

**Table 2. Mean number of co-authored reports by each resident per rotation.**

| (No. rotations) | Pre-pandemic (N) [CI] | Pandemic (N) [CI] | *p* |
|---|---|---|---|
| All (551) | 239.6 (369) [222.2–260.0] | 215.3 (182) [190.8–239.8] | .13 |
| *Readout model* | | | |
| Hybrid (385) | 214.5 (255) [195.2–233.8] | 209.6 (130) [180.3–238.9] | .78 |
| Remote (166) | 300.5 (114) [258.8–342.2] | 229.4 (52) [183.3–275.5] | .043* |
| *Division* | | | |
| AI (98) | 348.8 (65) [298.7–399.0] | 342.9 (33) [258.9–427.0] | .90 |
| Thoracic (75) | 229.6 (53) [180.1–279.2] | 171.8 (22) [120.0–223.7] | .17 |
| MSK (65) | 227.4 (44) [196.7–258.3] | 224.3 (21) [177.3–271.3] | .91 |
| Neuro (95) | 213.3 (62) [188.4–238.2] | 195.5 (33) [155.4–235.7] | .43 |
| NM (56) | 114.8 (37) [94.8–134.7] | 131 (19) [97.8–164.2] | .37 |
| Pediatric (91) | 362.1 (61) [300.0–424.3] | 271.6 (30) [202.8–340.5] | .08 |
| Women's (71) | 96.8 (47) [75.8–117.7] | 95.1 (24) [62.8–127.3] | .93 |

Two sided *t*-test

* denotes *p* values < .05. AI: abdominal imaging; Thoracic: thoracic imaging; MSK: musculoskeletal; Neuro: neuroradiology; NM: nuclear medicine; Pediatric: pediatric radiology.

resident per rotation, from 300.5 CI [258.8, 342.2] before the pandemic to 229.4 CI [183.3, 275.5] during the pandemic (Fig 1A). By contrast, a commensurate decrease was not observed for all combined rotations utilizing a hybrid model (212.4 CI [195.2, 233.8] to 209.6 CI [180.3, 238.9]; *p* = 0.81). When evaluating each division separately, during the pandemic, the mean number of studies interpreted by residents per rotation for the two remote divisions decreased by 25% for thoracic imaging (from 229.6 CI [180.1, 279.2] to 171.8 CI [120.0, 223.7]), and 25% for pediatric imaging (from 362.1 CI [300.0, 424.3] to 271.6 CI [202.8, 340.5]), but neither met the threshold for statistical significance when considered individually (*p* = .17 and *p* = .08, respectively) (Table 2). There was no such trend observed among any of the individual hybrid divisions. The frequency of imaging modalities within each rotation for pre-pandemic and pandemic periods is detailed in S2 Table.

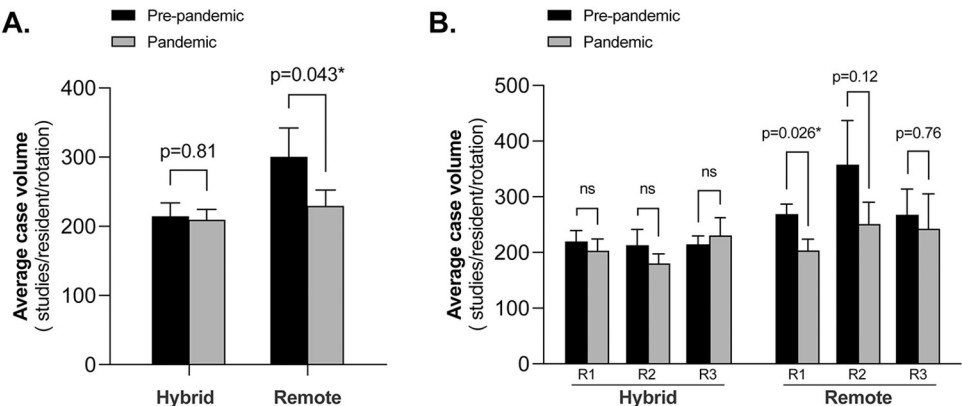

**Fig 1. Resident case volumes before and during the pandemic.** Readout models were adopted for all faculty in remote divisions and for a proportion of faculty in hybrid divisions during the pandemic period (grey column, July 1 to December 31, 2020). Resident co-authored reports were compared to pre-pandemic data (black column; July 1 to December 31, 2018, 2019). **A)** Average case volumes were normalized to rotation for all the residents (R1-R3, N = 56) and to **B)** resident by level.

## Resident case volumes per rotation in divisions employing a remote versus hybrid readout model, by level of training

A sub-analysis of the mean number of studies by level of training demonstrates several trends (Fig 1B, and S1 Table). As noted above, there was a significant decrease in the mean number of studies read by all resident levels combined (R1-R3) for rotations in remote divisions, but no corresponding decline was observed for hybrid rotations. There was a statistically significant decline in mean studies read by R1 residents for remote rotations (from 268.9 [232.6–305.2] to 203.5 [160.8–246.2]; $p$ = .026) but not for hybrid rotations (from 219.7 [180.5–258.9] to 203.8 [161.1–246.5]; $p$ = .65), nor for all rotations combined (from 234.9 [205.6–264.2] to 203.7 [172.1–235.3]; $p$ = .18). There was no statistically significant difference in the mean number of studies read by R2 residents from before the pandemic in either remote rotations (357.8 [278.9–436.7] to 251 [167.3–334.7]; $p$ = .12) or hybrid rotations (213.0 [184.6–241.3] to 180.7 [146.4–215.1]; $p$ = .18). However, there was a significant decline (from 262.0 [227.8–296.3] to 202.8 [168.0–237.6]; $p$ = .042) of the mean number of studies read by R2 residents when examining all rotations regardless of modality. By contrast, we found no significant differences in the average number of studies interpreted by R3 residents in remote, hybrid, or all combined rotations ($p$ = .44, $p$ = .76, $p$ = .74, respectively).

## Multivariate regression analysis for resident case volumes per rotation in remote divisions

We performed multivariate regression analyses to account for potential confounders affecting the number of studies interpreted by residents. As shown in Table 3, our regression analysis incorporated both a main specification utilizing continuous variables ranging from 0%-100% to account for the modality composition of each rotation as well as an alternative specification that instead controlled for innate differences in rotation characteristics to ensure durability. These generated similar estimates for the impact of the remote readout format on resident case volumes. When using controls to hold modality and division constant, respectively, we estimate that residents on remote rotations during the pandemic interpreted 82 fewer studies ($p$ = .005) and 74.9 fewer studies ($p$ = .011) with the default or alternative controls specified in Table 3 than they would have otherwise. By contrast, we did not observe a significant difference in pre-pandemic versus pandemic resident-interpreted studies completed in rotations that transitioned to hybrid readout models.

To better evaluate for a differential impact of remote readouts by trainee experience, we reran the main regression separately for each resident training level (regression results are reported in S3 Table). Despite the reduced sample size (since each regression had only about one-third as many observations), a statistically significant effect was seen for R1s in remote rotations, with the regression model for this cohort showing a net decrease of 72.3 studies interpreted ($p$ = .007). The effect was not significant for R2 or R3 residents ($p$ = .099 and $p$ = .29, respectively). Consistent with the prior regression results examining all residents at once, no significant changes were seen on hybrid rotations for any individual resident level.

Finally, to illustrate the effect of the transition to a remote model for each resident class, coefficients from the default model (Table 3) and 3 class-specific regression models (S3 Table) were used to calculate the marginal effects of the readout model on a standardized rotation holding other variables constant. Overall, residents on a hypothetical rotation would be expected to have had a −32% change on a remote rotation (252 to 170) but essentially no change (2.6%) on a hybrid rotation (227 to 233), as shown in Fig 2. Similarly, R1s, R2s, and R3s would be expected to have seen changes of −40% (180 to 108), −30% (341 to 240), and −26% (250 to 184) for studies interpreted in remote divisions, respectively. By contrast, case

**Table 3. Multivariate regression analysis.**

| Variable † | Default | Alternative Controls |
|---|---|---|
| R1 | reference | reference |
| R2 | 15.7 | 20.1 |
| R3 | 32.3* | 16.6 |
| Pre•Remote | reference | omitted |
| Pand•Remote | -81.6** | -74.9* |
| Pre•Hybrid | -25.1 | 3.9 |
| Pand•Hybrid | -19.2 | omitted |
| *Modality* | | |
| Radiography | reference | |
| CT/MR | -149.8*** | |
| Fluoro | 142.7 | |
| Mammo | -304.5*** | |
| NM | -187.7*** | |
| US | 110.3* | |
| *Division* | | |
| AI (98) | | reference |
| Thoracic (75) | | -109.2** |
| MSK (65) | | -118.1*** |
| Neuro (95) | | -139.7*** |
| NM (56) | | -226.8*** |
| Pediatric (91) | | 12.8 |
| Women's (71) | | -251.8*** |
| *Constant* | 311.5 | 332.1 |
| $R^2$ | 0.241 | 0.258 |
| *F* stat | <0.0001 | <0.0001 |

† Cross terms are denoted with • between interacting variables. *p* values of $< .05$, $< .01$, and $< .001$ are denoted with *, **, ***, respectively. Pre: pre-pandemic period; Pand: pandemic period between July 1st -Dec 2020. AI: abdominal imaging; Thoracic: thoracic imaging; MSK: musculoskeletal; Neuro: neuroradiology; NM: nuclear medicine; Pediatric: pediatric radiology, FL: Fluoro, MG: Mammography, XR: Radiographs, US: Ultrasound, NM: Nuclear Medicine, CT/MR: Cross-sectional studies

volumes for the same classes of residents rotating on hybrid rotations changed by −8% (197 to 181), −10% (242 to 219), and 5% (280 to 293), respectively.

## Discussion

Remote work was rapidly adopted at the onset of the COVID-19 pandemic across most industries and remains prevalent in academic radiology departments despite the relaxation of social distancing guidelines and the implementation of vaccines and treatments to mitigate the effects of the virus. While the initial impact of the pandemic on radiology resident education is well documented, there is limited evidence of the effect of an ongoing reliance on remote readouts on resident education and case volumes. In this study, we demonstrate the impact of remote readouts on radiology resident case volumes.

The pandemic period we analyzed (July to December 2020), was chosen specifically because institutional volumes had returned to normal during this time, enabling a controlled comparison of the divisions with a remote readout model to their pre-pandemic status; it also enabled a direct comparison to divisions resuming in-person readouts because two divisions—thoracic

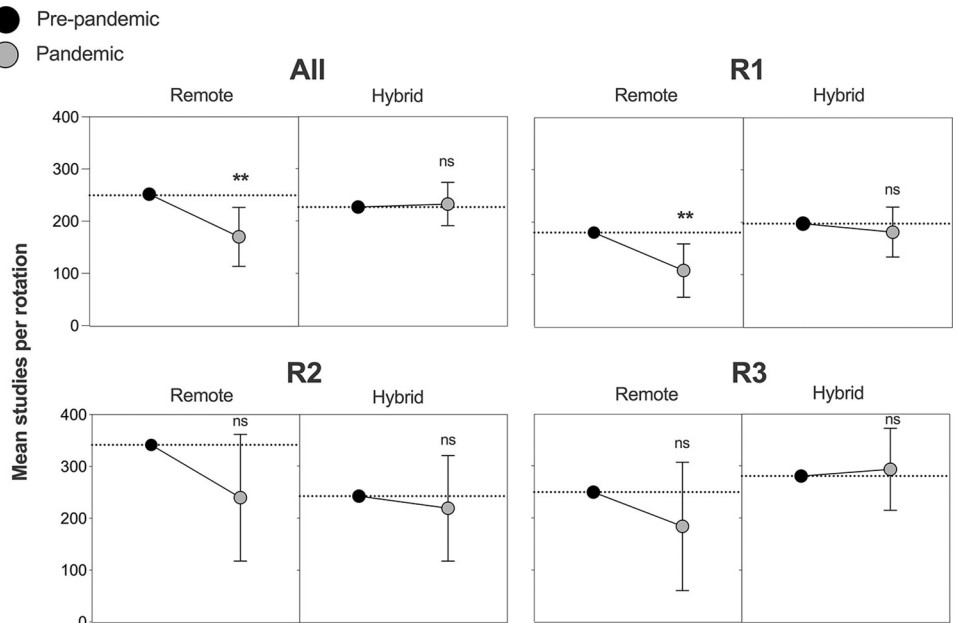

**Fig 2. The effect of the transition to a remote readout model on resident case volumes.** A model of multivariate regression analysis controlling for image modality was used to calculate the marginal effects of the readout model on resident case volumes before or during the pandemic. Results are shown for all residents and subdivided by resident level (R1-R3). The pre-pandemic period (shown in grey) encompasses the latter half of 2018 and 2019 and is compared to the pandemic period during latter half of 2020. $p$ values < .05 and < .01 are denoted by * and **, respectively; error bars represent CI for pandemic values.

and pediatric radiology—continued to operate with all non-procedural faculty interpreting remotely. The hybrid divisions resumed having at least one on-site attending always present in the reading room available to interact, teach, and review studies with residents. Residents in hybrid divisions interpreted the same average number of cases per rotation as in pre-pandemic years (212.4 to 209.6 studies per resident/rotation). In contrast, residents in remote divisions read considerably fewer (300.5 to 229.4 studies per resident/rotation). The trend was observed for both of the remote divisions but not for the hybrid divisions. These results were unlikely affected by differences in studies available to interpret because the total number read by the faculty of each division was similar to pre-pandemic levels. We performed multivariate analysis to account for potentially confounding variables such as the composition of imaging modalities read by each division and innate divisional differences (see Table 3 and S2 Table). Even after accounting for a possible shift in imaging modality distribution, a remote rotation in 2020 would be expected to encompass 82 fewer studies than an identical one prior ($p$ = .005), and a similar effect of 75 fewer studies was seen when explicitly controlling for differences innate to each division (Fig 2A). The regression analysis demonstrates a concerning decrease for a resident case volume in a purely remote rotation when compared to the preceding 2 years (171 vs 253 studies interpreted); such a large drop may reflect many missed educational opportunities with the potential for downstream negative effects on resident education. It is important to continue tracking residents affected by the pandemic to better understand if there are differences in board exam preparedness or clinical acumen, as resident case volumes are well described to correlate with clinical performance and board pass rate [11,12]. Although reduced radiology resident case volumes are well documented to accompany the drastically reduced institutional volumes during the COVID-19 pandemic, to our knowledge, this study represents the first dedicated analysis of the impact specifically of remote readouts. Our results

may partly explain the lagging return of resident volumes observed by Poyiadji et al. despite the normalization of faculty-only volumes [22].

A retrospective study of the subjective experiences of faculty and residents using home workstations found that, even though the residents did not perceive a difference in their own case volumes their faculty were more likely to perceive a decrease [20]. A study of residents working on-site or remotely on a home workstation by Freeman et al. reported no significant difference in case volume but the offsite residents were self-selected, and the composition of resident levels at each site was not reported and could have affected the outcome (19). In contrast to these studies, all our residents were on-site and in the reading room. Our data are supported by subjective evidence from surveys of both radiology faculty and trainees [15,21,23], with a consensus opinion that remote work negatively impacts education. In a survey conducted by Seghers et al., which compared the experiences of radiology faculty working remotely versus on-site, most respondents rated their ability to teach trainees as significantly decreased, with an average rating of 2.7 out of 10 on a scale where 0 represents the maximum diminishment of the work experience when working from home compared to in-hospital [15]. Similarly, the results of a recent survey by the American Alliance of Academic Chief Residents [21], could be considered a referendum on the efficacy of remote learning: a supermajority (74%) of chief residents reported that remote readouts were significantly less effective when conducted in-person. The majority of the chief residents also reported inferior learning in remote case conferences and didactics.

Junior trainees have also been shown to be likelier to report a negative effect of remote readouts on their education [23]. In addition to the main regression demonstrating an overall decrease in case volumes on remote rotations among all residents, our subanalysis examining each resident level in isolation provides additional evidence of a more pronounced effect amongst junior residents, with a predicted 40% decreased case volume for R1 residents (Fig 2B). Although the subanalyses for R2s and R3s did not show statistically significant decreases on remote rotations, this is potentially related to the low power of the study when subdivided to this degree, and the topic deserves future investigation. Residents who miss out on learning opportunities early in their training risk developing poor foundational skills, limiting their ability to build upon increasingly complex knowledge and skills. This could be accompanied by decreased confidence in their learning ability and their motivation for high achievement. Further studies to understand if remote readouts differentially impact junior residents are essential, as this could significantly affect their ability to progress in their training.

Remote readouts constitute a fundamental shift in workflow and introduce several factors that could affect resident productivity, warranting further investigation. Conventionally, radiology residents interpret exams independently, produce a preliminary report, and then review the studies alongside an attending physician at the same workstation. Traditionally, most of the teaching occurs during this time. However, the transition to remote readouts, where the attending and resident are physically separated, significantly alters this dynamic. Technical hurdles and challenges accompany the shift to a remote readout format, which has new software, hardware, or network requirements that introduce complexity to readouts and points of potential failure that could reduce resident efficiency. Furthermore, the absence of in-person supervision and collaboration may decrease resident motivation or raise issues in effective communication and teaching. For example, residents may find asking questions or clarifying points more challenging, leading to reduced confidence and slower study completion. These factors may particularly affect junior residents, as they may rely more on supervision, teaching, and feedback from attendings. Finally, attendings may find it easier or more efficient to simply interpret studies on their own, rather than conduct remote readouts.

There may also be unanticipated effects related to this fundamental change in workflow. One unexpected finding in our results illustrates that such a change may have unforeseen implications on resident education. At our institution, pediatric sonographers typically review the findings of each completed study with the radiologist via a dedicated on-site mobile phone. However, residents increasingly assumed this responsibility under the remote readout model as attendings worked remotely. Despite an overall decrease in resident volumes in the pediatric radiology division, the proportion of sonographic studies read by residents significantly increased, doubling from 9.9% to 18.4% (S2 Table, $p < 0.001$). This shift suggests that workflow changes could lead to imbalanced exposure to the variety of studies necessary for a complete resident experience.

Several factors may limit the generalizability of our findings. First, our data were acquired from a single institution. Second, the effect of implementing the readout models was evaluated for a relatively short period of six months during the pandemic. Unfortunately, the analysis could not be extended beyond this timeframe because of drastically decreased institutional volumes in earlier months and because the remote divisions transitioned to hybrid divisions afterward. No division retained a purely remote readout model for longer than 6 months; therefore, it is not possible with our results to determine whether the observed decreases in resident case volumes were transient. The effect may be less pronounced over time as residents become more accustomed to changes in workflow or learning format, and as the residency program responds to resident feedback. Over time, the negative impact on resident case volumes could lessen as technology, and new practice patterns are more familiar and effectively integrated. Furthermore, evaluating the difference between remote divisions to hybrid divisions is suboptimal because it is not a direct comparison between remote and in-person readouts, as remote readouts took place at some albeit much lower frequency in the hybrid divisions. This could mask the magnitude of effects attributed to remote readouts. While resident clinical case volumes are critical to resident education, there are other fundamental aspects of training physicians that we did not assess in our study, which the presence of onsite faculty could specifically promote. For example, in the remote setting, meaningful mentor-mentee camaraderie may be less common and mentorship less effective. Similarly, there may be poorer multidisciplinary exposure if fewer consultants visit the reading room and rather contact the offsite faculty directly. These factors may adversely affect the trainee's inspiration for novel research, professional ambitions, or attainment of career goals. Finally, a less personable approach to radiology education may negatively affect the mental health of residents and attendings and potentially increase the incidence of burnout.

The widespread adoption of remote radiology staffing as a response to the COVID-19 pandemic has become a paradigm in academic radiology departments despite limited research into potentially negative effects on the education and case volumes of radiology trainees. Our study provides evidence indeed that remote readouts may be detrimental to resident case volumes, potentially with greater consequence to junior residents. In light of these findings, programs should exercise caution to mitigate potential negative effects until further investigation occurs across multiple institutions and over a longer period. If difficult to avoid, remote readouts could be limited for the most junior residents. Continued surveillance by a relatively simple measure would be to document each resident co-authored report as either a remote or in-person readout, which would enable a direct comparison for future research.

In conclusion, our study provides evidence that residents interpret fewer studies when they are participating in only remote readouts. Academic radiology departments must remain vigilant for unintended consequences posed by an increasingly remote workforce, particularly for their most junior trainees.

## Supporting information

**S1 Table. Mean number of resident co-authored studies per rotation, by readout model and resident year.** Pre-pandemic encompasses July 1st through December 31st 2018 and 2019 and is compared to the pandemic period during July 1st through December 31st of 2020. Two-sided *t*-test, * denotes statistically significant p values<0.05.
(DOCX)

**S2 Table. Percent composition of imaging modalities for all rotations and the remote divisions.** †pre: the period before the pandemic encompasses July 1st through December 31st 2018 and 2019 and is compared to the pandemic period July 1st through December 31st 2020. Two-sided *t*-test, * denotes statistically significant p values<0.05. FL: Fluoro, MG: Mammography, XR: Radiographs, US: Ultrasound, NM: Nuclear Medicine, CT/MR: Cross-sectional studies.
(DOCX)

**S3 Table. Multivariate regression by resident level.** † Cross terms are denoted with • between interacting variables. *p* values of $< .05$, $< .01$, and $< .001$ are denoted with *,**,***, respectively. Pre: pre-pandemic period; Pand: pandemic period between July 1st and December 31st, 2020. FL: Fluoro; MG: Mammography; XR: Radiographs; US: Ultrasound, NM: Nuclear Medicine, CT/MR: Cross-sectional studies.
(DOCX)

## Author Contributions

**Conceptualization:** Emile B. Gordon, Peter Wingrove, Barton F. Branstetter IV.

**Data curation:** Emile B. Gordon, Peter Wingrove, Marion A. Hughes.

**Formal analysis:** Emile B. Gordon, Peter Wingrove, Barton F. Branstetter IV, Marion A. Hughes.

**Investigation:** Emile B. Gordon, Peter Wingrove, Barton F. Branstetter IV, Marion A. Hughes.

**Methodology:** Emile B. Gordon, Peter Wingrove, Barton F. Branstetter IV, Marion A. Hughes.

**Supervision:** Emile B. Gordon, Barton F. Branstetter IV, Marion A. Hughes.

**Validation:** Peter Wingrove, Marion A. Hughes.

**Writing – original draft:** Emile B. Gordon.

**Writing – review & editing:** Emile B. Gordon, Peter Wingrove, Barton F. Branstetter IV, Marion A. Hughes.

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
