## [Decision Letter · Decision Letter 0]

7 May 2023

PDIG-D-23-00148

Evidence for an adverse impact of remote readouts on radiology resident productivity: implications for training and clinical practice

PLOS Digital Health

Dear Dr. Gordon,

Thank you for submitting your manuscript to PLOS Digital Health. After careful consideration, we feel that it has merit but does not fully meet PLOS Digital Health's publication criteria as it currently stands. Therefore, we invite you to submit a revised version of the manuscript that addresses the points raised during the review process.

Please submit your revised manuscript within 60 days Jul 06 2023 11:59PM. If you will need more time than this to complete your revisions, please reply to this message or contact the journal office at digitalhealth@plos.org. Please include the following items when submitting your revised manuscript:

We look forward to receiving your revised manuscript.

Kind regards,

Haleh Ayatollahi

Section Editor

PLOS Digital Health

Journal Requirements:

1. We ask that a manuscript source file is provided at Revision. Please upload your manuscript file as a .doc, .docx, .rtf or .tex.

2. Please provide separate figure files in .tif or .eps format only and remove any figures embedded in your manuscript file. Please also ensure that all files are under our size limit of 10MB.

Additional Editor Comments (if provided):

The manuscript is interesting and well-written. Please consider the following points in your revision.

1- I assume the abstract should be unstructured and shorter based on the journal instructions.

2- In the abstract, please revise the aim of the study to make it clearer. It should be consistent with the aim written in the introduction section.

3- In the methods section, please use appropriate subheadings to explain research setting, participants, sampling, research instrument, data collection and analysis, ethics approval, etc. 

4- Please remove the footers which show different sections of the manuscript.

5- Please improve the readability of the manuscript to avoid any possible typos, unclear/ complex sentences, etc.

6- In the discussion section, please provide possible reasons for interpreting fewer studies by the radiology residents during remote rotations.

Reviewers' comments:

Reviewer's Responses to Questions

**Comments to the Author**

1. Does this manuscript meet PLOS Digital Health’s publication criteria? Is the manuscript technically sound, and do the data support the conclusions? The manuscript must describe methodologically and ethically rigorous research with conclusions that are appropriately drawn based on the data presented.

Reviewer #1: Yes

Reviewer #2: Partly

2. Has the statistical analysis been performed appropriately and rigorously?

Reviewer #1: Yes

Reviewer #2: Yes

3. Have the authors made all data underlying the findings in their manuscript fully available (please refer to the Data Availability Statement at the start of the manuscript PDF file)?

Reviewer #1: Yes

Reviewer #2: Yes

4. Is the manuscript presented in an intelligible fashion and written in standard English?

Reviewer #1: Yes

Reviewer #2: Yes

5. Review Comments to the Author

Reviewer #1: This study investigates the impact of hybrid vs remote reporting of radiology attendings on reporting numbers for residents. This is a well-constructed report and is of interest and relevance to radiology training programmes globally. There are a few points which should be addressed, major revisions advised. 

Major:

1. The authors have categorised the sub-specialty departments based on their remote vs hybrid practice during the pandemic, however it’s unclear what the practice was in these departments pre-pandemic. Was there a change in the extent of remote reporting in 2020? This is of particular relevance for the thoracic and paediatric departments – if the authors can demonstrate that there was a change in practice pre-pandemic vs pandemic, this would strengthen the link between the two and limit the possibility of other confounders accounting for the results. Also related to this, how was it determined where senior faculty were reporting from for each six month block?

2. The results are interesting, however the discussion lacks explanation as to why this drop in resident reporting might be. In practice, do residents normally report a study, then save the report/release a preliminary report to the clinicians until the attending reviews the report and finalises it with any necessary changes? Do they have to speak to the attending for each report? It would be useful to clarify how this works logistically in the authors’ department in the manuscript. If resident’s simply save/release preliminary reports and wait for them to be reviewed, why do the authors think the residents are reporting less in 2020 on remote rotations? 

Minor:

1. Consider adding radiology as a keyword. 

2. Although this is clarified later on in the manuscript, it is unclear in the abstract which group is working remotely. As it stands, it could be interpreted as residents working remotely instead of the attendings. 

3. I have been asked to comment on the availability of the raw data but I could not find/did not have access to this during my review.

4. Can the authors briefly explain what wRVUs are in the manuscript. Most non-US radiologists will not be familiar with this term. 

5. Why did the authors group CT and MR into one group instead of two? Also, I don’t think either of these terms were spelled out before acronyms were introduced. 

6. Results line 144: It’s confusing that the authors are investigating 2018-2020 but it mentions here that data was investigated over 18 months?

7. Results line 152: “Departmental wRVUs in 2020 (463,781) were 92% of the 2018 and 2019 average (477,179)”. 463781/477179 = 97% not 92%. 

8. Results line 159: Which half years of 2018 and 2019 are these figures referring to? In the following sentences it’s also unclear which halves of 2018, 2019 and 2020 are being compared. 

9. Can the authors confirm that there was no significant change in the attending staff of the departments which could serve as a confounder?

Reviewer #2: A good written manuscript showing the decrease in residence interpretation of studies at remote rotations. I believe expansion of some parts is necessary before proceeding to publication of the manuscript.

-an overview of the effect of telemedicine in general and remote health in the covid era is necessary in the introduction before introduction residence remote interpretation for a better flow of the manuscript. a good paper to select info from: 

 - Telemedicine in the COVID-19 Era: A Narrative Review Based on Current Evidence. Int J Environ Res Public Health. 2022 Apr 22;19(9):5101. doi: 10.3390/ijerph19095101;

-an explanation why the radiology residences and not other specialisation was selected for this study 

-and/or if a correlation with other specialisations in regards to tele-learning is of valid proposal for future studies. 

-if possible, a correlation of board pass rate and residence case volumes for this period should be introduced in the manuscripit if available.

6. PLOS authors have the option to publish the peer review history of their article (what does this mean?). If published, this will include your full peer review and any attached files.

**Do you want your identity to be public for this peer review?** For information about this choice, including consent withdrawal, please see our Privacy Policy.

Reviewer #1: No

Reviewer #2: Yes: NITTARI GIULIO

---

## [Decision Letter · Decision Letter 1]

18 Jul 2023

Evidence for an adverse impact of remote readouts on radiology resident productivity: implications for training and clinical practice

PDIG-D-23-00148R1

Dear Dr. Gordon,

We are pleased to inform you that your manuscript 'Evidence for an adverse impact of remote readouts on radiology resident productivity: implications for training and clinical practice' has been provisionally accepted for publication in PLOS Digital Health.

Best regards,

Haleh Ayatollahi

Section Editor

PLOS Digital Health

Reviewer Comments (if any, and for reference):

Reviewer's Responses to Questions

**Comments to the Author**

1. If the authors have adequately addressed your comments raised in a previous round of review and you feel that this manuscript is now acceptable for publication, you may indicate that here to bypass the “Comments to the Author” section, enter your conflict of interest statement in the “Confidential to Editor” section, and submit your "Accept" recommendation.

Reviewer #1: All comments have been addressed

Reviewer #2: All comments have been addressed

2. Does this manuscript meet PLOS Digital Health’s publication criteria? Is the manuscript technically sound, and do the data support the conclusions? The manuscript must describe methodologically and ethically rigorous research with conclusions that are appropriately drawn based on the data presented.

Reviewer #1: Yes

Reviewer #2: Yes

3. Has the statistical analysis been performed appropriately and rigorously?

Reviewer #1: Yes

Reviewer #2: Yes

4. Have the authors made all data underlying the findings in their manuscript fully available (please refer to the Data Availability Statement at the start of the manuscript PDF file)?

Reviewer #1: No

Reviewer #2: Yes

5. Is the manuscript presented in an intelligible fashion and written in standard English?

Reviewer #1: Yes

Reviewer #2: Yes

6. Review Comments to the Author

Reviewer #1: The authors have addressed my comments. The only remaining point is that the link they provided for data access yields an error message. Once this is fixed I recommend acceptance of the manuscript.

Reviewer #2: A valuable review work by the authors has been carried out. The authors have integrated all requested requests and corrections under review. This has greatly improved the quality of this manuscript which deals with an important topic of great relevance and interest for various scientific sectors. I believe that the proofreading that the authors carried out made the manuscript of excellent quality and therefore ready to be accepted for publication.

7. PLOS authors have the option to publish the peer review history of their article (what does this mean?). If published, this will include your full peer review and any attached files.

**Do you want your identity to be public for this peer review?** For information about this choice, including consent withdrawal, please see our Privacy Policy.

Reviewer #1: No

Reviewer #2: **Yes: **NITTARI GIULIO
